# Knowledge distillation in deep learning and its applications

Abdolmaged Alkhulaifi, Fahad Alsahli and Irfan Ahmad

Department of Information and Computer Science, King Fahd University of Petroleum and Minerals, Dhahran, Saudi Arabia



## ABSTRACT

Deep learning based models are relatively large, and it is hard to deploy such models on resource-limited devices such as mobile phones and embedded devices. One possible solution is knowledge distillation whereby a smaller model (student model) is trained by utilizing the information from a larger model (teacher model). In this paper, we present an outlook of knowledge distillation techniques applied to deep learning models. To compare the performances of different techniques, we propose a new metric called distillation metric which compares different knowledge distillation solutions based on models' sizes and accuracy scores. Based on the survey, some interesting conclusions are drawn and presented in this paper including the current challenges and possible research directions.

## INTRODUCTION

Deep learning has succeeded in several fields such as Computer Vision (CV) and Natural Language Processing (NLP). This is due to the fact that deep learning models are relatively large and could capture complex patterns and features in data. But, at the same time, large model sizes lead to difficulties in deploying them on end devices.

To solve this issue, researchers and practitioners have applied knowledge distillation on deep learning approaches for model compression. It should be emphasized that knowledge distillation is different from transfer learning. The goal of knowledge distillation is to provide smaller models that solve the same task as larger models (*Hinton, Vinyals & Dean, 2015*) (see Fig. 1); whereas, the goal of transfer learning is to reduce training time of models that solve a task similar to the task solved by some other model (cf. *Pan & Yang (2009)*). Knowledge distillation accomplishes its goal by altering loss functions of models being trained (student models) to account for output of hidden layers of pre-trained models (teacher models). On the other hand, transfer learning achieves its goal by initializing parameters of a model by learnt parameters of a pre-trained model.

There are many techniques presented in the literature for knowledge distillation. As a result, there is a need to summarize them so that researchers and practitioners could have a clear understanding of the techniques. Also, it is worth noting here that knowledge distillation is one of the ways to compress a larger model into a smaller model with comparable performance. Other techniques for model compression include row-rank factorization, parameter sharing, transferred/compact convolutional filters, and parameter

Corresponding author
Irfan Ahmad,
irfan.ahmad@kfupm.edu.sa

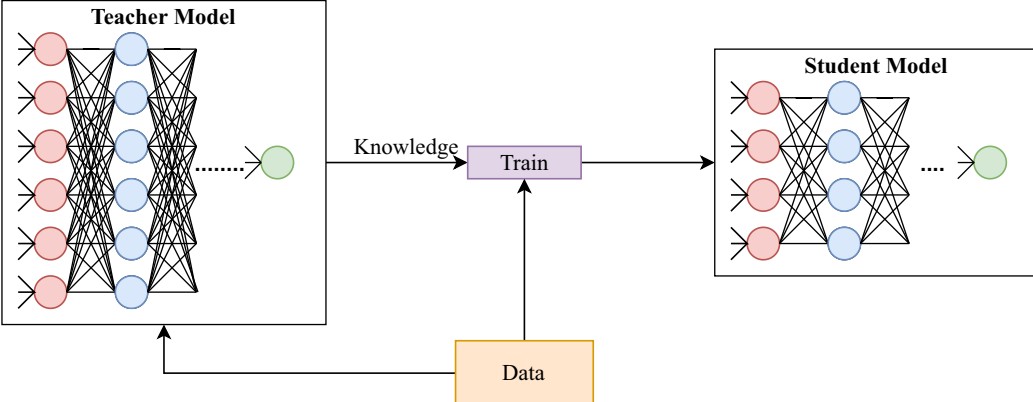

**Figure 1 A generic illustration of knowledge distillation.**

pruning as presented by *Cheng et al. (2017)*. To the best of our knowledge, there is no separate published survey on knowledge distillation techniques which motivated us to present a survey on recent knowledge distillation techniques for deep learning. Since there are many proposed knowledge distillation methods, we believe that they should be compared appropriately. Knowledge distillation approaches can be compared by several metrics such as reductions in model sizes, accuracy scores, processing times and so on. Our main criteria are reductions in model sizes and accuracy scores. Accordingly, we propose a metric–termed distillation metric–that takes into account the two criteria.

The main objectives of this work are to provide an outlook on the recent developments in knowledge distillations and to propose a metric for evaluating knowledge distillation approach in terms of reduction in size and performance. The paper also discusses some of the recent developments in the related field to understand the knowledge distillation process and the challenges that need to be addressed. The rest of the paper is organized as follows: In "Background", we provide a background on knowledge distillation. In "Distillation Metric", we present and discuss our proposed distillation metric. "Survey" contains the surveyed approaches and "Applications of Knowledge Distillation" contains some applications of knowledge distillation. We provide our discussion on surveyed approaches and an outlook on knowledge distillation in "Discussion and Outlook". Finally, we present our conclusions in "Conclusions".

## SURVEY METHODOLOGY

We searched papers on the topic of knowledge distillation in Google Scholar and selected the ones that were recent and not covered in previous similar surveys in the field. Papers published in the year 2017 and onward were included in the current work. Moreover, the papers were shortlisted based on the quality which was judged by the publication venue, i.e., indexed journals and relevant conferences such as International Conference on Machine Learning, Neural Information Processing Systems, AAAI Conference on Artificial Intelligence, International Conference on Learning Representations, conference on Computer Vision and Pattern Recognition, International Conference on Computer

Vision, Interspeech and the International Conference on Acoustics, Speech and Signal Processing. The papers were also selected based on their impact, i.e., citation count. Published works were searched using phrases containing the keywords such as "Knowledge Distillation", "Knowledge Distillation in Deep Learning" and "Model compression". Moreover, if a number of papers were retrieved in a specific topic, the papers that were published in less relevant journals and conferences or those having lower citation counts were excluded from the survey.

The available literature was broadly categorized into two sub-areas: techniques using only soft labels to directly train the student models and techniques using knowledge from intermediate layers to train the student models which may or may not use the soft labels. Accordingly, the survey was structured into two major sections each dealing with one of the broad categories. These sections were further divided into subsections for ease of readability and comprehensibility.

## BACKGROUND

Knowledge distillation was first introduced by *Hinton, Vinyals & Dean (2015)*. The main goal of knowledge distillation is to produce smaller models (student models) to solve the same task as larger models (teacher models) with the condition that the student model should perform better than the baseline model. Baseline models are similar to the student models but trained without the help of a teacher model. The distilling process can be achieved by using the soft labels, the probability distribution predicted by the teacher, in addition to the hard label which is represented as a one-hot vector, to train a student model. In this case, the student is trained with a loss function that minimizes the loss between its predictions and the hard and soft labels. Furthermore, one may distill the knowledge from the logits and feature maps of the teacher's intermediate layers. Logits are the output of a fully connected intermediate layer while feature maps are the output of a convolution layer. In this case, the loss function can be defined to minimize the differences between the selected intermediate layers of the corresponding teacher and student models. The feature extractor part of a network, i.e., the stack of convolution layers, are referred to as backbone. There are no conventions that guide student models' sizes. For example, two practitioners might have student models with different sizes although they use the same teacher model. This situation is caused by different requirements in different domains, e.g., maximum allowed model size on some device.

There exist some knowledge distillation methods that target teacher and student networks having the same size (e.g., *Yim et al. (2017)*). In such cases, the knowledge distillation process is referred to as self-distillation and its purpose is to further improve the performance by learning additional features that could be missing in the student model due to the random initialization *Allen-Zhu & Li (2020)*. Although an algorithm is developed to distill knowledge from a teacher model to a student model having the same size, the algorithm can be used to distill knowledge from a teacher to a smaller student, as well. This is because, based on our survey, there is no restriction on model sizes, and it is up to model designers to map teacher's activations to student's. So, in general settings, knowledge distillation is utilized to provide smaller student models that have

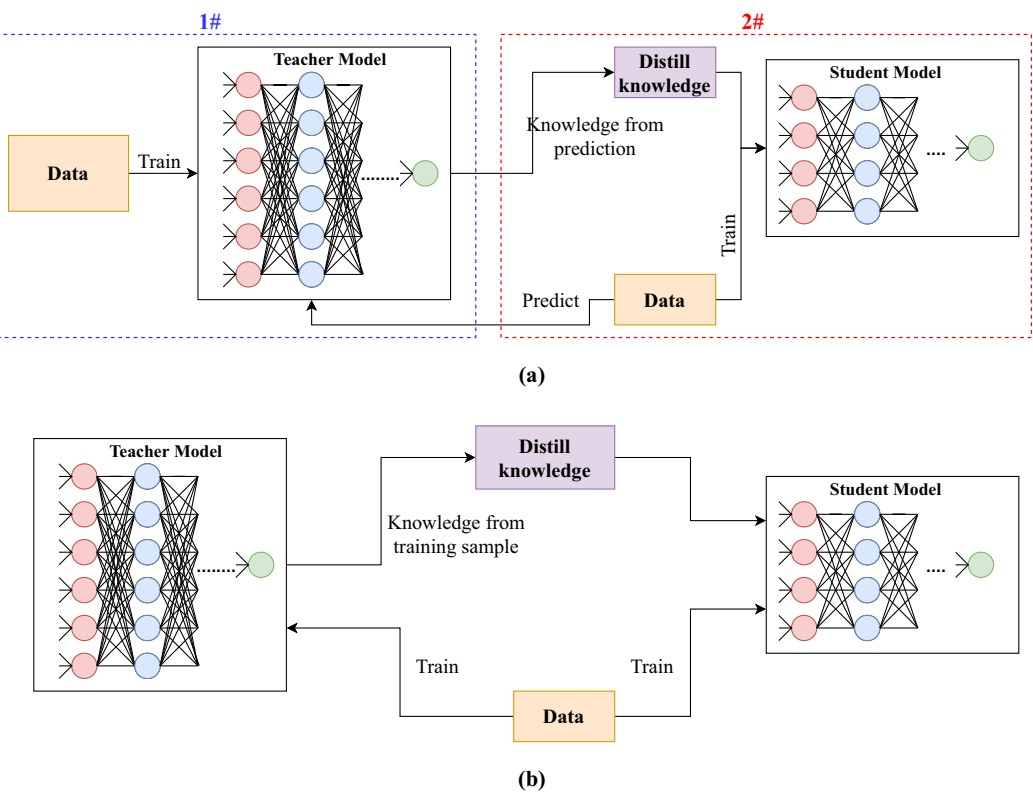

**Figure 2 Illustration of knowledge distillation using (A) pre-trained teacher model (offline) and (B) while training the teacher model simultaneously (online).**

comparable accuracy scores to their corresponding teacher models. The distillation process can be performed in offline or online manner. In offline distillation, the knowledge distillation process is performed using a pre-trained teacher model. While online distillation is for methods that perform knowledge distillation while training the teacher model. The illustration of the two subcategories can be seen in Fig. 2.

Consequently, one could compare different knowledge distillation algorithms by their reductions in model sizes. In addition, algorithms might be compared by how much accuracy they maintain compared to teacher models. There is no rule that governs how much reduction is best for all cases. For instance, if one needs to apply a knowledge distillation algorithm, they need to compare the algorithm's performance, in terms of reductions in size and accuracy, to their system's requirements. Based on the requirements, they can decide which algorithm fits best in their situation. To ease the process of comparison, we developed a distillation metric which can compare knowledge distillation results based on model sizes and accuracy scores. For more details on the distillation metric, please refer to "Distillation Metric".

There are different knowledge distillation approaches applied to deep learning models. For example, there exist approaches that distill knowledge from a single teacher to a single student. Other approaches distill knowledge from several teachers to a single student. Knowledge distillation could also be applied to provide an ensemble of student networks.

In "Survey", we present recent knowledge distillation approaches that are applied on deep learning based architectures.

## DISTILLATION METRIC

We propose distillation metric to compare different knowledge distillation methods and to select suitable model for deployment from a number of student models of various sizes. The metric incorporates ratio of a student's size to teacher's size and student's accuracy score to teacher's accuracy score. To have a good reduction in size, first ratio should be as small as possible. For a distillation method to have a good maintainability of accuracy, second ratio should be as close to 1 as possible. To satisfy these requirements, we develop the following equation:

$$DS = \alpha \times (student_s teacher_s) + (1-\alpha) \times (1-student_a teacher_a) \qquad (1)$$

where DS stands for distillation score, $student_s$ and $student_a$ are student size and accuracy respectively, and $teacher_s$ and $teacher_a$ are teacher size and accuracy respectively. Parameter $\alpha \in [0, 1]$ is a weight to indicate importance of first and second ratio, i.e., size and accuracy. The weight is assigned by distillation designers based on their system's requirements. For example, if some system's requirements prefer small model sizes over maintaining accuracy, designers might have $\alpha > 0.5$ that best satisfies their requirements.

It should be noted that when a student's accuracy is better than its teacher, the second ratio would be greater than 1. This causes the right operand of the addition operation (i.e., 1 − second ratio) to evaluate to a negative value. Hence, DS is decreased, and it could be less than zero especially if weight of the second ratio is larger. This is a valid result since it indicates a very small value for the first ratio as compared to the second ratio. In other words, this behaviour indicates a large reduction in model size while, at the same time, providing better accuracy scores than the teacher model. As presented in "Survey", a student model with a better accuracy is not a common case. It could be achieved, for example, by having an ensemble of student models.

Regarding the behaviour of the distillation metric, it is as follows: The closer the distillation score is to 0, the better the knowledge distillation. To illustrate, an optimal knowledge distillation algorithm would provide a value that is very close to 0 for the first ratio (e.g., the student's size is very small as compared to the teacher's size), and it would produce a value of 1 for second ratio (e.g., the student and the teacher models have same accuracy score). As a result, the distillation score approaches 0 as the first ratio approaches 0 and the second ratio approaches 1.

To demonstrate the usage of distillation metric, we use the results reported in *Walawalkar, Shen & Savvides (2020)* using CIFAR100 dataset *Krizhevsky (2009)* and the Resnet44 architecture *He et al. (2016)*. In their experiment, they trained four student models having relative sizes of 62.84%, 35.36%, 15.25% and 3.74% as compared to the teacher model. The teacher model achieved 71.76% accuracy, while the students achieved 69.12%, 67.04%, 62.87% and 43.11% accuracy, respectively. Considering that the model accuracy and size reductions are equally important, we set $\alpha = 0.5$. Calculating the distillations metric for the four student models we get a score of 0.333, 0.210, 0.138 and

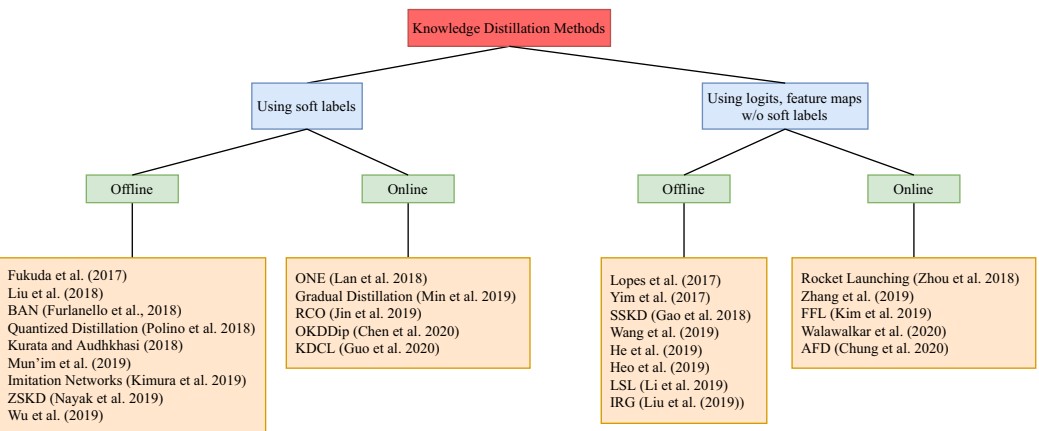

**Figure 3** A tree diagram illustrating the different knowledge distillation categories of methods and the different branches within each category.

0.218 respectively. Based on these results, we can notice that the model with the relative size of 15.25% (100,650 parameter) has the best balance between size and accuracy as compared to the teacher model and the other student models.

## SURVEY

This section includes recent work that targets knowledge distillation in deep learning. It is divided into two categories. The first category considers work that distills knowledge from the soft labels of the teacher model to train the student. Soft labels refers to the output of the teacher model. In case of classification tasks, the soft labels represent the probability distribution among the classes for an input sample. The second category, on the other hand, considers works that distill knowledge from other parts of the teacher model, optionally including the soft labels. Within each category, we further divide knowledge distillation methods into two subcategories: (1) offline distillation and (2) online distillation. A summary can be found in Fig. 3. In this survey, our main criteria are change in sizes and accuracy scores of student models against the corresponding teacher models. Regarding experiment results for the surveyed work, they are summarized in Tables 1 and 2.

### Techniques that distill knowledge from soft labels of the teacher models

#### *Offline distillation*

*Fukuda et al. (2017)* proposed a knowledge distillation approach by training a student model using multiple teacher models. Unlike other multi-teacher approaches that average the output of the teacher models to create the soft labels and then used to train the student model (*Wu, Chiu & Wu, 2019*; *Chebotar & Waters, 2016*; *Markov & Matsui, 2016*), the approach proposed by *Fukuda et al. (2017)* was to opt out of combining the teachers output distribution and to train the student on the individual output distribution. The authors argued that this would help the student model to observe the input data from different angles and would help the model to generalize better.

**Table 1 Summary of knowledge distillation approaches that utilize soft labels of teacher to train student model.** In case of several students, results of student with largest size reduction are reported. In case of several datasets, dataset associated with the lowest accuracy reduction is recorded. Baseline models have the same size as the corresponding student models, but they were trained without the teacher models.

| Reference | Targeted architecture | Utilized data | Reduction in accuracy compared to teacher | Improvement in accuracy compared to baseline | Reduction in size |
|---|---|---|---|---|---|
| Offline distillation | | | | | |
| *Fukuda et al. (2017)* | CNN | Aurora (*Hirsch & Pearce, 2000*) | 0.782% | 2.238% | – |
| *Liu, Wang & Matwin (2018)* | Decision tree | MNIST (*LeCun, 1998*) | 12.796% | 1-5% | – |
| *Furlanello et al. (2018)* | DenseNet (*Huang et al., 2017*) | CIFAR-100 (*Krizhevsky, 2009*) | 2.369% (increase) | – | – |
| *Polino, Pascanu & Alistarh (2018)* | Wide ResNet (*Zagoruyko & Komodakis, 2016*) | CIFAR-100 | 0.1813% | – | 52.87% |
| *Kurata & Audhkhasi (2018)* | LSTM | SWB Switchboard subset from HUB5 dataset (https://catalog.ldc.upenn.edu/LDC2002T43) | 2.655% | – | 55.07% |
| *Mun'im, Inoue & Shinoda (2019)* | Seq2Seq | WSJ Wall Street Journal dataset (https://catalog.ldc.upenn.edu/LDC93S6B) | 8.264% | 8.97% | 89.88% |
| *Kimura et al. (2019)* | CNN | MNIST | 10.526% (increase) | 16.359% | – |
| *Nayak et al. (2019)* | CNN | MNIST | 0.57% | – | 40% |
| *Wu, Chiu & Wu (2019)* | ResNet (*He et al., 2016*) | HMDB51 (*Kuehne et al., 2011*) | 0.6193% | – | 58.31% |
| Online distillation | | | | | |
| *Lan, Zhu & Gong (2018)* | ResNet | CIFAR100, | – | 6.64% | – |
| *Min et al. (2019)* | Micro CNN | Synthetic Aperture Radar Images Synthetic Aperture Radar Images dataset (https://www.sandia.gov/radar/imagery/index.html) | 0.607% | – | 99.44% |
| *Jin et al. (2019)* | MobileNetV2 (*Sandler et al., 2018*) | ImageNet (*Deng et al., 2009*) | 9.644% | 6.246% | 70.66% |
| *Chen et al. (2020)* | ResNet | CIFAR100, | – | 5.39% | – |
| *Guo et al. (2020)* | ResNet | CIFAR100, | 1.59% | 6.29% | 34.29% |

While deep learning has achieved great success across a wide range of domains, it remains difficult to identify the reasoning behind model predictions, especially if models are complex. To tackle this issue, *Liu, Wang & Matwin (2018)* proposed a method of converting deep neural networks to decision trees via knowledge distillation. The proposed approach consisted of training a Convolutional Neural Network (CNN) first with the given dataset. Using the feature set from the training dataset as input and the logits from the trained model as output, they trained a classification and regression trees model, where logits are scores before the Softmax activations.

**Table 2 Summary of knowledge distillation approaches that distills knowledge from parts other than or in addition to the soft labels of the teacher models to be used for training the student models.** In case of several students, results of student with largest size reduction are reported. In case of several datasets, dataset associated with the lowest accuracy reduction is recorded. Baseline models have the same size as the corresponding student models, but they were trained without the teacher models.

| Reference | Targeted architecture | Utilized data | Reduction in accuracy compared to teacher | Improvement in accuracy compared to baseline | Reduction in size |
|---|---|---|---|---|---|
| Offline distillation | | | | | |
| Lopes, Fenu & Starner (2017) | CNN | MNIST | 4.8% | 5.699% (decrease) | 50% |
| Yim et al. (2017) | ResNet | CIFAR-10 | 0.3043% (increase) | – | – |
| Gao et al. (2018) | ResNet | CIFAR-100 | 2.889% | 7.813% | 96.20% |
| Wang et al. (2019) | U-Net | Janelia (Peng et al., 2015) | – | – | 78.99% |
| He et al. (2019) | MobileNetV2 | PASCAL (Everingham et al., 2010) | 4.868% (mIOU) | – | 92.13% |
| Heo et al. (2019) | WRN | ImageNet to MIT scene (Quattoni & Torralba, 2009), | 6.191% (increase) | 14.123% | 70.66% |
| Li et al. (2019) | CNN | UIUC-Sports (Li et al., 2010) | 7.431% | 16.89% | 95.86% |
| Liu et al. (2019) | ResNet | CIFAR10 | 0.831% | 2.637% | 73.59% |
| Online distillation | | | | | |
| Zhou et al. (2018) | WRN | CIFAR-10 | 1.006% | 1.37% | 66% |
| Zhang et al. (2019) | ResNet18 | CIFAR100 | 13.72% | – | – |
| Kim et al. (2019) | CNN | CIFAR100 | 5.869% | – | – |
| Walawalkar, Shen & Savvides (2020) | ResNet | CIFAR10 | 1.019% | 1.095% | 96.36% |
| Chung et al. (2020) | WRN | CIFAR100 | 1.557% | 6.768% | 53.333% |

*Furlanello et al. (2018)* proposed an ensemble knowledge distillation method called Born-Again Neural Networks. The method considered the issue of teacher and student models having the same architecture (self distillation). The method first trained a teacher model using a standard approach. Then, it trained a student model using the ground truth and teacher's predictions. After that, it trained a second student model using the ground truth and previous student's predictions and so on (see Fig. 4). For instance, student$_i$ was trained by utilizing training labels and predictions of student$_{i-1}$ for $i \in [1, n]$, where $n$ is the number of student models. When student models were used for prediction, their results were averaged. *Furlanello et al. (2018)* claimed that the method would produce better models since it was based on ensemble learning, and a model was trained on training labels and predictions of a previously trained model.

*Polino, Pascanu & Alistarh (2018)* developed a knowledge distillation approach for quantized models. Quantized models are models whose weights are represented by a limited number of bits such as 2-bit or 4-bit integers. Quantized models are used to develop hardware implementations of deep learning architectures as they provide lower power consumption and lower processing times compared to normal models (full-precision models) (*Courbariaux, Bengio & David, 2015*). The distillation approach had two variants. First variant was called quantized distillation, and it trained a quantized student model and a full-precision student model. The two models were trained according

to true labels and the teacher's predictions. The main purpose of the full-precision model was to compute gradients and update the quantized model accordingly. As claimed by *Polino, Pascanu & Alistarh (2018)* the reason behind this process was that there was no objective function that accounted for quantized weights. This issue motivated *Polino, Pascanu & Alistarh (2018)* to develop the second variant of their knowledge distillation approach, and they called it differentiable quantization. They defined an objective function to address the issue of quantized weights. As a result, there would be no need for a full-precision student model.

*Kurata & Audhkhasi (2018)* developed a distillation approach that targeted sequence models (*Bahdanau et al., 2016*) for speech recognition. The distillation goal was to transfer knowledge of a Bidirectional Long Short-Term Memory model to an LSTM model. This was achieved by considering teacher's soft labels and comparing outputs of three time steps of the teacher network to a single time step output of the student network. Furthermore, *Mun'im, Inoue & Shinoda (2019)* proposed a distillation approach for Seq2Seq speech recognition. The approach trained a student network to match teacher k-best outputs generated with beam search, where $k$ is a hyper-parameter.

When tackling problems where only few samples are available, it can make models overfit easily. *Kimura et al. (2019)* proposed a method that allowed training networks with few samples while avoiding overfitting using knowledge distillation. In their approach, they first trained a reference model with few samples using Gaussian processes (GP) instead of neural networks. Then, the samples used for training were augmented using inducing point method via iterative optimization. Finally, the student model was trained with the augmented data using loss function defined in the paper with the GP teacher model to be imitated by the student model. *Nayak et al. (2019)* proposed a method to train the student model without using any dataset or metadata. The method worked by extracting data from the teacher model through modeling the data distribution in the softmax space. Hence, new samples could be synthesized from the extracted information and used to train the student model. Unlike generative adversarial networks where they generate data that is similar to the real data (by fooling a discriminative network), here the synthesized data was generated based on triggering the activation of the neurons before the softmax function.

*Wu, Chiu & Wu (2019)* developed a multi-teacher distillation framework for action recognition. Knowledge was transferred to the student by taking a weighted average of three teachers soft labels (see Fig. 4). The three teachers are fed different inputs. The first teacher is fed with the residual frame, while the second teacher is fed with motion vector. The last teacher is fed with the I-frame image, similar to the student model.

### Online distillation

In *Lan, Zhu & Gong (2018)*, the authors proposed the On-the-fly Native Ensemble (ONE) knowledge distillation. ONE takes a single model and creates multiple branches where each branch can be considered as individual models. All the models share the same backbone layers. The ensemble of models is viewed as the teacher while a single branch is selected to be the student model. During training, the model is trained with three loss

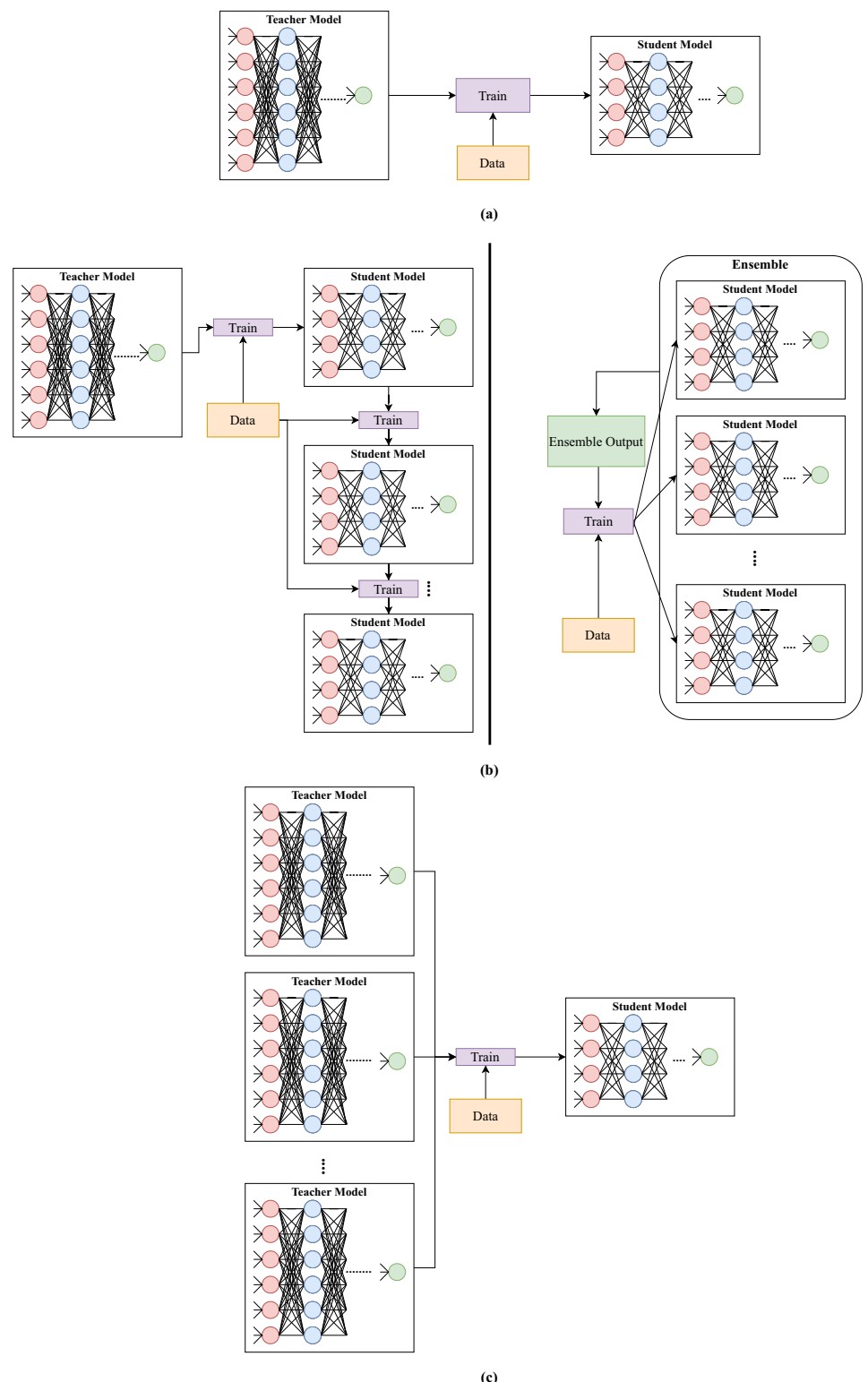

**Figure 4 Illustration of different types of knowledge distillation depending on the number of teachers and students.** (A) Knowledge distillation from one teacher to one student. (B) Knowledge distillation from one teacher to multiple students. (C) Knowledge distillation from multiple teachers to one student.

functions. The first loss function is the cross entropy between the predictions of each individual branch and the ground truth. The second loss function is the cross entropy between the prediction distribution of the ensemble of all models and the ground truth. The third loss function is the Kullback Leibler divergence between the prediction distribution of the whole ensemble and the individual branches. The prediction distribution of the ensemble of models is produced using a gating mechanism.

*Min et al. (2019)* presented a technique called gradual distillation arguing that quantized distillation indirectly results in loss of accuracy and it is difficult to train directly from the hard and soft labels. The gradual distillation approach trains the teacher model and the student model simultaneously. The output from the teacher's network at each step is used to guide the student learning. Accordingly, the loss function for the student's network has two components: the cross-entropy loss between the output of the student's network and the hard labels, and the cross-entropy loss between the student output and the teacher's target.

Training a compact student network to mimic a well-trained and converged teacher model can be challenging. The same rationality can be found in school-curriculum, where students at early stages are taught easy courses and further increase the difficulty as they approach later stages. From this observation, *Jin et al. (2019)* proposed that instead of training student models to mimic converged teacher models, student models were trained on different checkpoints of teacher models until teacher models converged. For selecting checkpoints, a greedy search strategy was proposed that finds efficient checkpoints that are easy for the student to learn. Once checkpoints were selected, a student model's parameters were optimized sequentially across checkpoints, while splitting data used for training across the different stages depending on its hardness defined by a hardness metric that was proposed by the authors.

An ensemble knowledge distillation approach named Online Knowledge Distillation with Diverse peers (OKDDip) was proposed by *Chen et al. (2020)*. OKDDip uses an ensemble of models as a teacher (named auxiliary peer) and a single model within the group as a student (named group leader). Unlike ONE, the ensemble of models can be independent models or have shared layers. Each model is trained to reduce the cross entropy between its predictions and the ground truth. Additionally, each model will take a weighted average of predictions of all models in the ensemble and uses Kullback Leibler divergence loss function between its prediction distribution and the weighted average of predictions of the ensemble. Each auxiliary peer will assign different weights to all other auxiliary peers in the group to determine how the prediction distribution is aggregated. For the group leader, it will just take the average of the prediction of all the auxiliary peers. The weight assignment process for the auxiliary peers takes the feature extracted for each peer and projects it to two sub-spaces by applying linear transformation with learned weights. The weights for each peer is then calculated similar to the self-attention mechanism using the projected sub-spaces *Vaswani et al. (2017)*.

Another ensemble knowledge distillation method was proposed by *Guo et al. (2020)* named Knowledge Distillation via Collaborative Learning (KDCL). KDCL trains on input data that is distorted differently for each student in the ensemble. The cross-entropy

loss function between prediction and hard labels is used to train each student model in addition to the Kullback Leibler divergence loss between the prediction and the soft labels. The authors proposed four different methods to generate the soft labels. The first method selects a single student's probability distribution in the ensemble as soft label that produces the minimum cross entropy loss. The second method finds the best linear combination of the students logtis that minimizes the cross-entropy loss through convex optimization and use it to generate the soft labels via softmax function. The third method subtracts the logit that corresponds to the target class from all logits for each student. Then, it constructs the ensemble logits by selecting the minimum logit for each class from all the students in the ensemble which later is fed to softmax to create the soft labels. The fourth method of producing the soft labels takes the weighted average of students' outputs. The weight for each student is assigned after every training epoch and it is based on its performance on the validation set.

Table 1 provides a summary of the presented work. It shows that the best achieved reduction in size is by *Min et al. (2019)* with a reduction of 99.44% in the number of parameters. We can also observe from the table that the best approach in terms of maintaining accuracy is proposed by *Kimura et al. (2019)* with an increase in accuracy by 10.526%. However, their work utilizes knowledge distillation to overcome overfitting when dealing with small amount of training samples. Furthermore, they used a Gaussian process as a teacher model which can explain the increase in accuracy of the student CNN model. Additionally, *Kimura et al. (2019)* approach helped the student model to generalize better on small number of training samples and achieve the highest increase in accuracy compared to the baseline model which overfitted on the training data.

## Techniques that sistills knowledge from other parts of the teacher model with or without soft labels

### Offline distillation

*Lopes, Fenu & Starner (2017)* proposed transferring knowledge to a student model using a metadata which holds a summary of activations of the teacher model during training on the original dataset, instead of using the original dataset used to train the teacher. The metadata includes top layer activation statistics, all layers activation statistics, all-layers spectral activation record, and layer-pairs spectral activation record. Then using one of the collected metadata, we can capture the view of the teacher model of the dataset and hence we can reconstruct a new dataset that can be used to train a compact student model. *Yim et al. (2017)* proposed a two-stage distillation for CNNs. The first stage defines two matrices between the activations of two non-consecutive layers. The first matrix corresponded to the teacher network, and the second matrix corresponded to the student network. Then, the student was trained to mimic the teacher's matrix. After that, the second stage began by training the student normally.

*Gao et al. (2018)* proposed to only train the backbone of a student model to mimic the feature extraction output of a teacher model. After that, the student model is trained on ground truth data while freezing the parameters of the backbone layers. The knowledge distillation process only happened during training of the backbone layers of the smaller

student model, which allowed it to be trained on different dataset than the teacher model. *Wang et al. (2019)* proposed a distillation method for encoder-decoder networks that trained a student model by comparing its soft labels to the teacher's labels and the ground truth. Moreover, the student will also compare its encoders outputs to that of the teacher.

*He et al. (2019)* proposed to train an auto-encoder network to compress feature maps of the teacher. The student is later trained to match the compressed feature maps of the teacher model. Additionally, the student was also trained to match its feature map affinity matrix to the of the teacher model. This was needed because student network could not capture long-term dependencies due to its relatively small size.

Unlike other knowledge distillation methods where neuron responses of teacher model is the focus when transferring knowledge to students, *Heo et al. (2019)* proposed to focus on transferring activation boundaries of teacher instead. Activation boundary is a hyperplane that decides whether the neurons are active or not. In *Pan & Srikumar (2016)*, decision boundary of neural network classifier was proven to be a combination of activation boundaries, which made them an important knowledge to be transferred to the student model. Based on this, *Heo et al. (2019)* proposed an activation transfer loss that penalized when neurons activations of teacher and student were different in the hidden layers. Since both the teacher and the student model, most likely, would not have the same number of neurons, the authors utilized a connector function that converts the vector of neurons of the student model to be the same size as the vector of neurons in the teacher model. By applying the proposed loss function, activation boundaries of the teacher model were transferred to the student model.

*Li et al. (2019)* introduced the Layer Selectivity Learning (LSL) framework for knowledge distillation. In LSL framework, some intermediate layers are selected in both the teacher and the student network. The selection process is done by feeding data to the teacher model and calculating the inter-layer Gram matrix and the layer inter-class Gram matrix using the feature vectors in order to find layers that are the most informative and discriminative across the different classes. The selection process can be applied to the student model by training it on a dataset alone in order to select the same number of intermediate layers. Once intermediate layers are selected from both the networks and aligned, the student network is trained with an alignment loss function, in addition to a loss function that minimizes the prediction loss, that minimizes the difference between the feature vectors of pairs of intermediate layers from the teacher and the student network. The alignment loss function will force the student's intermediate layers to mimic the intermediate layers of the teacher model. Since the feature vectors of a pair of intermediate layers of the teacher and student network will not have the same dimensions, the feature vector is fed to a fully-connected layer that projects the feature vectors to the same dimensions.

Previous knowledge distillation approaches only considered the instance features (the soft output of the layer) to be transferred from the teacher model to the student model. This made it hard for student models to learn the relationship between the instance feature and the sample with a different and compact model architecture. *Liu et al. (2019)*

proposed representing the knowledge using an instance relation graph (IRG). For each layer in the model, an IRG was created where vertices represent the instance features and edges represent the instance relationship. Transformation function was defined to transform two IRG of adjacent layers into new IRG which contained the feature-space knowledge of the two layers. Using IRG of the teacher layers and the student layers, a loss function was defined to help train the student model using the knowledge encapsulated in the IRG of the teacher.

### Online distillation

*Zhou et al. (2018)* proposed to train the teacher (named booster net) and the student (named lightweight net) together. This was done by sharing the backbone layers of the two models during training and then using a function where it contained the loss of the booster network, the loss of the lightweight network, and the mean square error between the logits before softmax activation of both the networks. To prevent the objective function from hindering the performance of the booster network, a gradient block scheme was developed to prevent the booster network's specific parameter from updating during the backpropagation of the objective function which would allow the booster network to directly learn from the ground truth labels. To improve their approach further, the authors used the knowledge distillation loss function from *Hinton, Vinyals & Dean (2015)* in their objective function.

*Zhang et al. (2019)* proposed an online self-distillation method that trains a single model. The model convolution layers is first divided into sections, where a branch is added after each shallow section that contains a bottleneck layer *He et al. (2016)*, fully connected layer and a classifier. The added branches are only used during training and it will let each section act as a classifier. The deepest classifier (original classifier after the last convolution layer) is considered the teacher model. The deepest classifier and each shallow classifier is trained using cross entropy between its prediction and the hard labels. Additionally, each shallow classifier is trained using Kullback Leibler divergence loss to minimize between its prediction and the soft labels of the deepest classifier. Moreover, each shallow classifier is trained using L2 loss between the feature maps of the deepest classifier and the feature maps of the bottleneck layer of each of the shallow classifiers.

*Kim et al. (2019)* proposed a learning framework termed Feature Fusion Learning (FFL) that can also acts as a knowledge distillation framework. An ensemble of models with either similar or different architecture is used in addition to a special model called fusion classifier. If FFL is used for knowledge distillation, we can consider any single individual model in the ensemble as a student model while the whole ensemble and the fusion classifier will act as the teacher. Each model in the ensemble is trained normally with the ground-truth labels while the fusion classifier takes the feature maps of all the models in the ensemble as an input in addition to the ground-truth labels. Furthermore, the ensemble models will distill its knowledge to the fusion classifier in the form of the average of all predictions to be used with Kullback Leibler divergence loss in order to transfer the knowledge of the ensemble to the fusion classifier. Moreover, the fusion classifier will

also distill its knowledge back to each model in the ensemble in the form of its prediction distribution to be used with Kullback Leibler divergence loss. This way, the knowledge distillation is mutual between the fusion classifier and the ensemble. After training, any model in the ensemble can be selected to be deployed or the whole ensemble with the fusion classifier can be deployed in case of lenient hardware constraints.

*Walawalkar, Shen & Savvides (2020)* proposed to train an ensemble of models that is broken down into four blocks, where all models share the first block of layers. The first model in the ensemble is considered the teacher (termed pseudo teacher in the paper). For each successive model (students), the number of channels in their convolution layers is reduced by an increasing ratio to the teacher model. During deployment, any model in the ensemble can be selected depending on the hardware constraints or, in cases of lenient constraints, the whole ensemble can be deployed. In addition to training each model using cross entropy between predictions and ground truth, an intermediate loss function is used to distill the knowledge of the intermediate block of layers (feature maps) of the teacher model to each of the student models. Moreover, Kullback Leibler divergence loss is used between the model prediction and the average predictions of the whole ensemble. Since the number of channels of the student models and the teacher model is not the same, an adaptation layer ($1 \times 1$ convolution) is used to map the student channels to the teacher channels. The intermediate loss function is a mean squared error between the feature maps of the teacher and student pair.

*Chung et al. (2020)* proposed online Adversarial Feature map Distillation (AFD) that trains two networks to mimic each other's feature maps through adversarial loss. Aside from training using the cross-entropy loss on the ground truth and Kullback Leibler divergence loss between the logits of the two networks, AFD trains a discriminator for each network that distinguishes between the feature map produced by the accompanying network and the other network. Each network in AFD is trained to fool its corresponding discriminator and minimize the adversarial loss. This in-turn will let the model learn the feature map distribution of the other network. In case of training two networks, one can be considered as the student (model with less parameters) and the other as the teacher model (with more parameters) and both the student and the teacher model will learn from each other. Due to the difference in the number of channels of the feature maps between the two networks, a transfer layer is used to convert the number of channels of the student network to that of the teacher network.

Table 2 provides a summary of the presented works. It shows that the best approach in terms of size reduction is proposed by *Li et al. (2019)* with a reduction of 95.86% in size. The table also shows that the best approach in terms of maintaining accuracy is proposed by *Heo et al. (2019)* with an increase in accuracy of 6.191%. However, their experiments were conducted on a teacher model that is trained and evaluated on two different datasets. Their experiments focused on combining knowledge transfer with knowledge distillation. As for the improvement compared to the baseline model, the LSL proposed by *Li et al. (2019)* achieved the best improvement of 16.89% increase in accuracy.

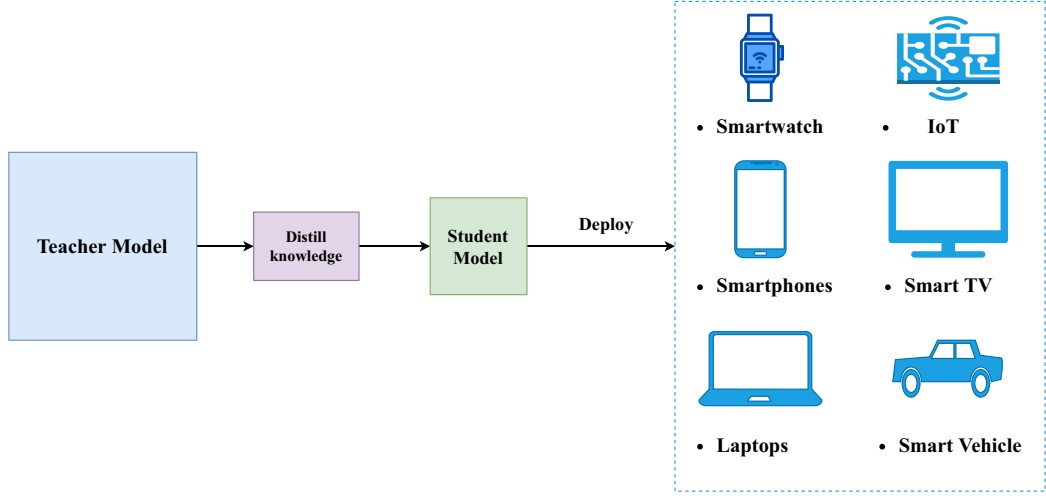

**Figure 5 Use cases for knowledge distillation to deploy deep learning models on small devices with limited resources.**

# APPLICATIONS OF KNOWLEDGE DISTILLATION

Traditionally, deep learning models typically run on Cloud computing platforms delivering the results to the smart devices over a network. Although this model is feasible in some situations, it is not preferred in many other situations where delay is not tolerable or data privacy is a concern. Moreover, unpredictable network connections between the cloud and the device can also pose significant challenges. Thus, running the deep learning system on local devices is an important requirement in many domains and has a wide variety of applications including smart cities, self-driving cars, smart homes, medical devices and entertainment *Véstias et al. (2020)*. Knowledge distillation allows developers to shrink down the size of deep learning models in order for them to fit into resource-limited devices having limited memory and power as illustrated in Fig. 5. In this section we present some typical applications of knowledge distillation based on the recent literature.

In *Chen et al. (2019)*, knowledge distillation was used to train a lightweight model for pedestrian detection which will enable fast pedestrian detection in smart vehicles with autonomous driving functionality. *Janveja et al. (2020)* presented a smartphone-based system for detecting driver fatigue based on frequency of yawning and frequency of eye closure. *Yang et al. (2018)* presented the use of MobileNets in addition to Batch Normalization and Swish activation function (cf. *Ramachandran, Zoph & Le (2017)*) to estimate the steering angle for self-driving cars.

In the domain of healthcare, *Esteva et al. (2017)* presented an end-to-end deep CNN based system to classify different types of skin cancer from skin images. The paper proposed the idea of deploying the system on smart phones so that a large population can easily access the diagnostic services. *Ahn et al. (2018)* presented a CNN based deep learning system to assist in capsule endoscopy. The idea is to adaptively control the capsule's image capturing frequency and quality based on detecting damaged areas in a patient's small intestine. To adaptively control the capsule moving through a patient's

intestine, the authors suggest pairing the capsule with an external device attached to the patient's waist which can process the incoming images in real-time and direct the capsule in terms of image frequency and quality. The authors identified some of the challenges that need to be addressed in order for the system to be of practical use. Among the challenges identified were the need for the system to have low latency and be efficient in battery usage. This can be achieved in part by developing light-weight models using knowledge distillation techniques.

*Plötz & Guan (2018)* presented the use of deep learning system trained on the cloud to be deployed on smart phones for human activity recognition (HAR) using the data available from smartphone sensors. The authors identified the challenge of dealing with resource constraints on these mobile devices and the use of knowledge distillation techniques to address some of these challenges. *Czuszynski et al. (2018)* presented hand-gesture recognition using recurrent neural networks (RNNs) deployed on smartphones. The idea of HAR based on spatio-temporal features from IoT devices like a cup, a toothbrush and a fork was presented in *Lopez Medina et al. (2019)*. Knowledge distillation was also used for training a small model for image classification which will help IoT-based security systems to detect intrusion (*Wang et al. (2020)*).

*Lane, Georgiev & Qendro (2015)* presented an audio-sensing deep learning framework for smartphones which can infer a number of situations such as the current environment (voice, music, water and traffic), stress detection, emotion recognition (anger, fear, neutral, sadness and happiness), and speaker identification using a smartphone's audio input. *Mathur et al. (2017)* presented a wearable vision system powered by deep learning that can process the camera images in real-time locally in the device for tasks such as face recognition, scene recognition, object detection, age and gender assessment from the face images, and emotion detection. Another work on object recognition on smartphones using deep learning systems was presented by *Fang, Zeng & Zhang (2018)*. *Chauhan et al. (2018)* presented a RNN based deep learning system for user authentication using breathing-based acoustics data. The trained system is evaluated on smartphones, smartwatches and Raspberry Pi. The authors show that model compression can help reduce the memory size by a factor of five without any significant loss in accuracy.

## DISCUSSION AND OUTLOOK

The distillation score proposed in this work can not be used as a fair comparison between the different methods mentioned in this work. Each reported method utilizes different datasets, architectures and uses knowledge distillation for different applications. *Blalock et al. (2020)* discussed the difficulty of assessing the state-of-the-art in model pruning as a model compression technique. The authors also listed various reasons why it is difficult to compare different pruning techniques including the ambiguities related to the architecture used or the metrics used to report the result. The authors also presented a list of best practices and proposed an open source library as a benchmark to standardize the experiments and evaluations.

Reporting the reduction in model size as well as change in accuracy for a student model as compared to the corresponding teacher model is useful in our opinion. Although most

authors report this information, some authors do not report either of the two pieces of information. Moreover, comparing the performance of a student model to a baseline model (e.g., trained-from-scratch model of comparable size to the student model) is also very informative, and we believe that it should be reported by authors.

Regarding the future of knowledge distillation, most researchers did not provide comments. Nevertheless, *Polino, Pascanu & Alistarh (2018)* suggested the use of reinforcement learning to enhance development of student models. According to the authors, it is not clear how to develop student models that meet memory and processing time constraints. Building a program based on reinforcement learning such that its objective is to optimize memory and processing time requirements would ease development of student models.

In addition, most researchers focus on CV tasks. For instance, out of the surveyed work, few considered NLP tasks. Recently, several language models based on transformer architecture (*Vaswani et al., 2017*) have been proposed such as Bidirectional Encoder Representations from Transformers (*Devlin et al., 2018*). These models have parameters in the order of hundreds of millions. This issue has motivated several researchers to utilize knowledge distillation (*Sanh et al., 2019*; *Sun et al., 2019*). However, knowledge distillation has not been well investigated yet. Transformer based language models provide better results, in terms of accuracy scores and processing times as compared to RNNs (*Devlin et al., 2018*; *Radford et al., 2019*). As a result, it is important to study knowledge distillation on such models so that relatively small and high performance models could be developed.

The idea that knowledge distillation is a one-way approach of improving the performance of a student model utilizing a teacher model has led some researchers (e.g., *Wang, Wang & Gao, 2018*; *Chung et al., 2020*; *Kim et al., 2019*) to explore other collaborative learning strategies where learning is mutual between teachers and students.

Based on some recent works such as *Hooker et al. (2019)* and *Hooker et al. (2020)*, measures like top-1 and top-5 accuracy masks some of the pitfalls of model compression techniques. The impact of model compression on true generalization capability of the compressed models are hidden by reporting models' overall performances using such measures. In general, difficult-to-classify samples are the ones which are more prone to under-perform on the compressed models. Thus, it seems that the systems' bias get further amplified which can be a major concern in many sensitive domains where these technologies will eventually be deployed such as healthcare and hiring. In addition, compressed models are less robust to changes in data. Addressing these concerns will be an important research direction in the area of model compression including knowledge distillation. One implication of the work is to report class-level performances instead of comparing one overall performance measure for the system such as accuracy. Macro-averaged $F1$ scores across all the classes may be a more useful performance measure than accuracy. Other appropriate measures need to be used for evaluation which can compare fairness and bias across the models. The authors presented two such measures in their work. Furthermore, it will be important to investigate these issues on more domains as the current papers looked mainly on the image classification problems. One approach that

might mitigate the above mentioned problems is to use a modified loss function during the distillation process that penalizes label misalignment between the teacher and the student models (e.g. *Joseph et al. (2020)*).

*Allen-Zhu & Li (2020)*, in a recent paper, argues how knowledge distillation in neural networks works fundamentally different as compared to the traditional random feature mappings. The authors put forward the idea of 'multiple views' of a concept in the sense that neural network, with its hierarchical learning, learns multiple aspects about a class. Some or all of these concepts are available in a given class sample. A distilled model is forced to learn most of these concepts from a teacher model using the soft labels or other intermediate representations during the distillation process. In addition, the student model learns its own concepts due to its random initialization. Now, in order to explain the findings of *Hooker et al. (2019)* and *Hooker et al. (2020)*, it seems that some of the less prevalent concepts which were learnt by the teacher model are missed by the student model which gives rise to increased biases in the student model.

## CONCLUSIONS

We present several different knowledge distillation methods applied on deep learning architectures. Some of the methods produce more than 80% decrease in model sizes (*He et al., 2019*; *Li et al., 2019*). Some other methods provide around 50% size reductions, but they maintain accuracy scores of teacher models (*Polino, Pascanu & Alistarh, 2018*; *Gao et al., 2018*). In addition, there exist distillation approaches that result in student models with better accuracy scores than their corresponding teacher models (*Heo et al., 2019*; *Furlanello et al., 2018*). Our criteria in the present study are based on reductions in models' sizes and accuracy scores. Consequently, we propose distillation metric which helps in comparing between multiple students of various sizes. We also highlight different contexts and objectives of some of the knowledge distillation methods such as limited or absence of the original dataset, improving interpretability, and combining transfer learning with knowledge distillation.

Moreover, knowledge distillation is a creative process. There are no rules that guide development of student models or mapping teacher's activations to student's although there have been some recent attempts to understand them in a deeper way. As a consequence, knowledge distillation highly depends on the domain where it is applied on. Based on requirements of the specific domain, model designers could develop their distillation. We advise designers to focus on simple distillation methods (or build a simpler version of some method) that target a relatively small number of student and teacher layers. This is an important step as it decreases the time needed for designers to get familiar with different behaviour of different distillation methods in their domain. After that, they could proceed with more complex methods as they would have developed intuitions about how the methods would behave on their domain of application. As a result, they could eliminate some methods without having to try them. In addition, designers could utilize distillation metric to assess their evaluations. Moreover, other relevant measures should be used in evaluating a technique and using the accuracy measure may not be sufficient by itself. Some of the challenges in the area were discussed in this paper in

addition to possible future directions. Last but not the least, we also discussed in this paper some of the practical applications of knowledge distillation in real-world problems.

### Funding
This work was supported by King Fahd University of Petroleum and Minerals (KFUPM), Dhahran, Saudi Arabia. No specific funding was received for this work. The funders had no role in study design, data collection and analysis, decision to publish, or preparation of the manuscript.

### Grant Disclosures
The following grant information was disclosed by the authors:
King Fahd University of Petroleum and Minerals (KFUPM), Dhahran, Saudi Arabia.

### Competing Interests
Irfan Ahmad is an Academic Editor for PeerJ Computer Science.

### Author Contributions
- Abdolmaged Alkhulaifi conceived and designed the experiments, performed the experiments, analyzed the data, performed the computation work, prepared figures and/or tables, authored or reviewed drafts of the paper, and approved the final draft.
- Fahad Alsahli conceived and designed the experiments, performed the experiments, analyzed the data, performed the computation work, prepared figures and/or tables, authored or reviewed drafts of the paper, and approved the final draft.
- Irfan Ahmad conceived and designed the experiments, performed the experiments, analyzed the data, authored or reviewed drafts of the paper, and approved the final draft.

### Data Availability
This is a survey paper and does not make use of data or code apart from the published articles.

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
