# Peer review of "Knowledge distillation in deep learning and its applications"

_PeerJ Computer Science, doi:10.7717/peerj-cs.474_

## Round 0.1 · original submission · Major Revisions

Pay special attention to the topics related to the experimental part of the paper and the validity of findings.

Reviewer 1 ·

Basic reporting

This review studies techniques which transfer the knowledge acquired by large deep learning models to smaller models, which can then be used in embedded and mobile devices.

The level of English is more than adequate, explanations are clear and accessible to a broad range of readers.

It mostly respects the PeerJ standard structure, with a few extra sections that makes sense for the contents of the overview. However, readers may appreciate some more subdivisions in the Survey section, which only has one subheading and one sub-subheading (I believe those should be at the same level instead, this may be a formatting errata). The acknowledgements section includes funders.

The review applies and is accessible to any deep learning practitioner, including those who may not be specialized in the topic but may want to embed a certain level of intelligent behavior in a small device, a situation where knowledge distillation techniques are of interest. This field has been reviewed recently but none of those reviews are published on a peer-reviewed journal as of now, so this would apparently be the first review of the topic in a reliable source, since the topic itself is also very recent.

The introduction of the manuscript introduces the concepts appropriately, but I think it is missing some examples as to what tasks can be achieved with deep learning in embedded/mobile devices (e.g. fitness tracking, sensor data compression?), since the main justification for knowledge distillation is the need of smaller deep learning models, but there is no explanation for what problems these models may solve.

Experimental design

The content of the article is well within the aims and scope of PeerJ Computer Science. The described methodology in order to collect studies and results seems appropriate and rigorous. It is also systematic, since it introduces an objective metric for the fitness of different algorithms to the problem, which takes into account the reduction in size as well as the preservation (or even improvement) of accuracy. The value of this metric is lower as performance in both aspects improves. The metric is relative to the sizes and accuracies of the models, and does not directly depend on the data used, but it is computed using the metrics reported by the original papers, so I am unsure about its ability to compare those models. The authors could justify briefly the level to which this metric is independent of the datasets used.

The survey seems diverse and comprehensive, all methods are sufficiently described and the explanations are put together well, including detailed information about the experiments and results of each study. There is, however, little to no visual aid to complement the textual explanations. I think a simple diagram outlining the main components of a deep learning-based knowledge distillation model (i.e. teacher, student, the flow of data and weights, or how the student is trained) would be very helpful to give the reader an intuition on what all these proposals have in common.

Cited sources are reliable in general, either from reputable journals and conferences or, at least, well-known papers on ArXiv.

Validity of the findings

The discussion of the results is sound, and several guidelines are provided on how to improve works in the topic. Some possible future directions are also mentioned and appropriately cited. The conclusions summarize the manuscript correctly and attempt to guide the novel reader on how to use these models.

Additional comments

In summary, my overall opinion of this paper is very good, but I believe some improvements could be made that would make it easier to read and comprehend. My suggestions are as follows: to extend the introduction with applications of deep learning in embedded devices, better subdivisions of the Survey section, and a diagram or two explaining the common points of the inner workings of these models.

Reviewer 2 ·

Basic reporting

The introduction could include some additional sentences to explain the main contributions and findings of the survey.

Section numbers in the last paragraph of the introduction do not appear.

For a survey, the background could be more formal, introducing key concepts and definitions. The authors could also detail the categories or perspectives for the survey analysis, such as inputs, algorithms, distillation/compression approaches, outputs, etc.

The title mentions “applications”, so I would expect a specific section summarizing current applications and others the authors suggest could be explored in the future. Some information is given, e.g., used datasets in each paper, but a section from the application perspective could be more practical for readers interested in particular domains.

Some specific sentences that authors should clarify are:
- Section “survey”, line 136. Did the authors exclude papers not presenting evaluation metrics or they were only discarded from the comparison?
- Section “survey”, line 142. Every neural network -> every deep neural network

Experimental design

The survey methodology to search, select and summarize the papers should be improved. The authors only use one source (Google Scholar), so many relevant papers could be missing. It is not clear if the search strings are independent or not. The number of papers found, filtered out and finally selected should be also indicated. Usually, exclusion and inclusion criteria are established to clearly state the reasons why papers are discarded and selected. Current quality criteria seem pretty subjective, i.e., which are minimum acceptable citation count, or which are the “relevant” journals and conferences. All this information is necessary for replicability.

Reporting of each paper is quite complete, but it is not easy to understand how the authors have organized the paragraphs of each category (soft labels, transformation). Both sections are large to read, so the authors could think if a subdivision would fit, e.g., based on the application, specificity (agnostic or depend on network architecture), purpose of the knowledge distillation process…

Validity of the findings

The authors propose a metric to compare knowledge distillation techniques, but it is not evaluated for any of the surveyed techniques. Having a new metric could be very useful for researchers and adding a short study showing how it is computed and interpreted for a subset of techniques would add value to the paper.

The authors compare and discuss the distillation scores obtained by different techniques as reported in the original publications. However, it is not clear if all these techniques are comparable, i.e, do they comprise the same input deep learning? I guess not, so averaging or comparing achieved reduction and accuracy improvement is a bit risky. The authors could try to extract some common behaviors among techniques depending on the targeted architecture, dataset/application, etc.

Additional comments

None.

---

## Round 0.2 · Minor Revisions

Reviewers appreciate the improvements done in the paper, and so do I.

Reviewer 1 ·

Basic reporting

This overview paper studies techniques which transfer the knowledge acquired by large deep learning models to smaller models, which can then be used in embedded and mobile devices.

The level of English is adequate, apart from some minor grammatical errata that should be edited out: e.g. "The main objectives of this work" -> "The main objective of this work"; "Also, the paper discuss" -> "The paper also discusses"; "deep learning models use to run" -> "deep learning models are used to run" (or just "typically run").

The introduction and background sections have been sufficiently improved.

Experimental design

This study fits within the scope of the journal and there is no recent peer-reviewed review of the topic, to my knowledge. The overall structure of the paper has been improved and is now easier to follow. Diagrams have been added to complement method descriptions.

Validity of the findings

no comment

Additional comments

Overall, the paper has received many improvements and the contents are now well organized and present the whole picture notably better. The only modification I would recommend is a quick revision for grammar mistakes such as those I marked above, for greater clarity. Apart from that, I would consider that this overview reaches publication quality.

Reviewer 2 ·

Basic reporting

As suggested, the authors have extended the background to include some of the concepts used along the paper. I would also move the definition of online and offline distillation (including figure 2) that now appears at the beginning of section 5 (lines 173-176) to the background.

The authors have also included a new section summarizing the applications, as requested. This section contributes making the paper more complete.

New figures are quite useful to understand the background concepts and the categories used to classify the papers.

The manuscript still contains some grammar mistakes (a few are listed below), so proof-reading is highly recommended before publication.

- Section 1: The main objectives of this works is => are
- Section1: Also, the paper discuss => discusses
- Section 3: It’s purposes => its purposes
- Section 5: the two sub-category => subcategories
- Section 6: deep learning models use to run => are usually run ?
- Section 6: To be practically in use => To be of practical use
- Section 6: To be low latency => To have low latency

Experimental design

I am not fully satisfied with the answer given by the authors about the survey methodology. Even if they do not want to conduct a systematic literature search, the process followed to find and select the papers should be better explained in the manuscript. It seems that the survey is focused on recent works not included in previous surveys, so the covered period of time should be given. The names of the journals and conferences considered as “relevant”, as well as the minimum citation count, should be reported as well. Even though these criteria might not be valid for a systematic review, the reader has the right to know how the authors choose papers. Otherwise, the “overview” of the area is strongly biased by the authors’ interest on certain papers, but the reader is not aware of it.

The new organization of the survey section has greatly contributed readability.

Validity of the findings

The authors have successfully addressed my comments about the validity of findings.

Additional comments

None.

---

## Author Rebuttal · Round 0.2

# Response to the Reviewers' Comments

We thank the editor and the reviewers for their efforts and for providing constructive comments and feedback. We believe that the modification of the original manuscript to address those remarks has significantly improved the manuscript. Following are the highlights of the changes done in the revised manuscript. The detailed reply to reviewers' comments follows:

1. Added a total of 8 figures grouped under 5 main figure captions.
2. Restructured the manuscript based on the reviewers' feedback including addition of subsections under the survey sections.
3. Added a new section titled "Applications of Knowledge Distillation in Deep Learning".
4. Enhanced the section "Discussion and Outlook" to reflect on the recent advances in the field.
5. Added a total of 15 new papers to include recent works in the area in addition to those needed to address the reviewers' comments.

**Reply to Reviewer #1:**

| | |
|---|---|
| This review studies techniques which transfer the knowledge acquired by large deep learning models to smaller models, which can then be used in embedded and mobile devices.<br><br>The level of English is more than adequate, explanations are clear and accessible to a broad range of readers. | We thank the reviewer for his/her efforts towards reviewing the manuscript and providing valuable feedback. |
| It mostly respects the PeerJ standard structure, with a few extra sections that makes sense for the contents of the overview. However, readers may appreciate some more subdivisions in the Survey section, which only has one subheading and one sub-subheading (I believe those should be at the same level instead, this may be a formatting errata). The acknowledgements section includes funders. | Based on the feedback from the reviewer, we have restructured the manuscript and have added sub-divisions under the "Survey" section. Now, the survey section is divided in two main sub-sections and each sub-section is further divided into two sub sections. We have also fixed the numbering issues in the updated manuscript. |
| The review applies and is accessible to any deep learning practitioner, including those who may not be specialized in the topic but may want to embed a certain level of intelligent behavior in a small device, a situation where knowledge distillation techniques are of interest. This | We would like to thank the reviewer for his/her efforts towards reviewing the manuscript and providing encouraging feedback.<br><br>We have created a new section titled "Applications of Knowledge Distillation in Deep Learning" where we discuss examples of the |

| | |
|---|---|
| field has been reviewed recently but none of those reviews are published on a peer-reviewed journal as of now, so this would apparently be the first review of the topic in a reliable source, since the topic itself is also very recent.<br><br>The introduction of the manuscript introduces the concepts appropriately, but I think it is missing some examples as to what tasks can be achieved with deep learning in embedded/mobile devices (e.g. fitness tracking, sensor data compression?), since the main justification for knowledge distillation is the need of smaller deep learning models, but there is no explanation for what problems these models may solve | tasks and problems that can be solved with knowledge distillation techniques. |
| The content of the article is well within the aims and scope of PeerJ Computer Science. The described methodology in order to collect studies and results seems appropriate and rigorous. It is also systematic, since it introduces an objective metric for the fitness of different algorithms to the problem, which takes into account the reduction in size as well as the preservation (or even improvement) of accuracy. The value of this metric is lower as performance in both aspects improves. The metric is relative to the sizes and accuracies of the models, and does not directly depend on the data used, but it is computed using the metrics reported by the original papers, so I am unsure about its ability to compare those models. The authors could justify briefly the level to which this metric is independent of the datasets used. | The metric is independent of the datasets used as it computes the relative reduction in the model size and the relative change in the accuracy. We agree with the reviewer that comparing two different systems performing different tasks may not be completely fair. The distillation metric may be more useful to compare different sub-solutions for a given task and select the best compromise between compression and accuracy.<br><br>We have added more clarification in the updated manuscript on the interpretation of the results from the metric and have removed the direct comparisons of different works based on the metric. |
| The survey seems diverse and comprehensive, all methods are sufficiently described and the explanations are put together well, including detailed information about the experiments and results of each study. There is, however, little to no visual aid to complement the textual explanations. I | We thank the reviewer for pointing this out. As suggested by the reviewer, we have added a total of 8 figures in the updated manuscript grouped into 5 main figures captions. |

| | |
|---|---|
| think a simple diagram outlining the main components of a deep learning-based knowledge distillation model (i.e. teacher, student, the flow of data and weights, or how the student is trained) would be very helpful to give the reader an intuition on what all these proposals have in common. | |
| The discussion of the results is sound, and several guidelines are provided on how to improve works in the topic. Some possible future directions are also mentioned and appropriately cited. The conclusions summarize the manuscript correctly and attempt to guide the novel reader on how to use these models.<br><br>In summary, my overall opinion of this paper is very good, but I believe some improvements could be made that would make it easier to read and comprehend. My suggestions are as follows: to extend the introduction with applications of deep learning in embedded devices, better subdivisions of the Survey section, and a diagram or two explaining the common points of the inner workings of these models | We thank the reviewer for his/her efforts towards reviewing the manuscript and providing encouraging feedback.<br><br>We have updated the manuscript based on the suggestions as detailed above. |

**Reply to Reviewer #2:**

| | |
|---|---|
| The introduction could include some additional sentences to explain the main contributions and findings of the survey. | We thank the reviewer for his/her efforts towards reviewing the manuscript and providing valuable feedback.<br>We have updated the introduction to add explanations about the main contributions and findings of the survey. |
| Section numbers in the last paragraph of the introduction do not appear. | We thank the reviewer for pointing this out.<br>We have fixed this issue in the updated manuscript. |
| For a survey, the background could be more formal, introducing key concepts and definitions. The authors could also detail the categories or perspectives for the survey analysis, such as inputs, algorithms, distillation/compression approaches, outputs, etc. | We have updated the background section to address the concerns of the reviewer by introducing the key concepts and definitions in addition to the categories or perspectives for the survey analysis. |

| | |
|---|---|
| The title mentions "applications", so I would expect a specific section summarizing current applications and others the authors suggest could be explored in the future. Some information is given, e.g., used datasets in each paper, but a section from the application perspective could be more practical for readers interested in particular domains. | We have created a new section titled "Applications of Knowledge Distillation in Deep Learning" where we discuss examples of the tasks and problems that can be solved with knowledge distillation techniques based on the recent publications. |
| Some specific sentences that authors should clarify are:<br>- Section "survey", line 136. Did the authors exclude papers not presenting evaluation metrics or they were only discarded from the comparison?<br>- Section "survey", line 142. Every neural network -> every deep neural network | We have clarified the mentioned sentences in the updated manuscript.<br>We did not exclude papers that do not present the evaluation metrics. We have removed the direct comparisons between different work using our proposed distillation metric. |
| The survey methodology to search, select and summarize the papers should be improved. The authors only use one source (Google Scholar), so many relevant papers could be missing. It is not clear if the search strings are independent or not. The number of papers found, filtered out and finally selected should be also indicated. Usually, exclusion and inclusion criteria are established to clearly state the reasons why papers are discarded and selected. Current quality criteria seem pretty subjective, i.e., which are minimum acceptable citation count, or which are the "relevant" journals and conferences. All this information is necessary for replicability. | We have selected papers from 2016 and beyond to include works that were not covered by previous surveys. Google Scholar indexes multiple sources including IEEE Explore, ScienceDirect, and Springer. Most of the scientific papers in the area are covered by these sources. We understand reviewer's concern that the current selection criteria are not completely objective as done in a systematic literature review or a mapping study.<br>We would like to point that the current paper is not a comprehensive survey on the topic but an outlook for the readers to have an overall introduction to the topic and we have selected representative works which covers the different ideas within the topic. We have clarified this in the updated manuscript and there is no mention of phrases like "system literature survey", 'systematic mapping study", and "comprehensive survey". |
| Reporting of each paper is quite complete, but it is not easy to understand how the authors have organized the paragraphs of each category (soft labels, transformation). Both sections are large to read, so the authors could think if a subdivision would fit, e.g., based on the application, specificity (agnostic or depend on network architecture), | We thank the reviewer for his/her efforts towards reviewing the manuscript and providing encouraging feedback.<br><br>Based on the feedback of the reviewer, we have restructured the survey. We have divided the major sections in the Survey into further subsections. We hope that the new organization is both easy for the readers and more meaningful, at the same time. We have also |

| purpose of the knowledge distillation process… | added some figures to further clarify the ideas using illustrations. |
|---|---|
| The authors propose a metric to compare knowledge distillation techniques, but it is not evaluated for any of the surveyed techniques. Having a new metric could be very useful for researchers and adding a short study showing how it is computed and interpreted for a subset of techniques would add value to the paper. | Based on the feedback from the reviewer, we have added a short study under the section "Distillation Metric" which shows how the metric can be computed and the results be utilized in the updated manuscript, |
| The authors compare and discuss the distillation scores obtained by different techniques as reported in the original publications. However, it is not clear if all these techniques are comparable, i.e, do they comprise the same input deep learning? I guess not, so averaging or comparing achieved reduction and accuracy improvement is a bit risky. The authors could try to extract some common behaviors among techniques depending on the targeted architecture, dataset/application, etc. | We thank the reviewer for pointing this out. We agree with the reviewer that comparing two different systems performing different tasks may not be completely fair. The distillation metric may be more useful to compare different sub-solutions for a given task and select the best compromise between compression and accuracy or when two different works are using the same initial architecture and the task. We have added more clarification in the updated manuscript on the interpretation of the results from the metric and have removed the direct comparisons of different works based on the metric. Also, the issues of comparing different distillation techniques have been discussed under the "Discussion and Outlook" section. |

---

## Round 0.3 · accepted · Accept

The paper is ready for publication. No more modifications are required

---

## Author Rebuttal · Round 0.3

## Response to the Reviewers' Comments

We would like to thank the editor and the reviewers for their efforts and for providing constructive comments and feedback throughout the review process. We believe that the modification of the manuscript to address those remarks has further improved it. Detailed replies to reviewers' comments follows:

**Reply to Reviewer #1:**

| | |
|---|---|
| This overview paper studies techniques which transfer the knowledge acquired by large deep learning models to smaller models, which can then be used in embedded and mobile devices. | We thank the reviewer for his/her efforts towards reviewing the manuscript and providing valuable feedback. |
| The level of English is adequate, apart from some minor grammatical errata that should be edited out: e.g. "The main objectives of this work" -> "The main objective of this work"; "Also, the paper discuss" -> "The paper also discusses"; "deep learning models use to run" -> "deep learning models are used to run" (or just "typically run"). | We thank the reviewer for pointing these errors. We fixed them in the updated manuscript. In addition, we proofread the manuscript to fix grammar mistakes elsewhere, as well. |
| This study fits within the scope of the journal and there is no recent peer-reviewed review of the topic, to my knowledge. The overall structure of the paper has been improved and is now easier to follow. Diagrams have been added to complement method descriptions. Overall, the paper has received many improvements and the contents are now well organized and present the whole picture notably better. The only modification I would recommend is a quick revision for grammar mistakes such as those I marked above, for greater clarity. Apart from that, I would consider that this overview reaches publication quality. | We would like to thank the reviewer for his/her efforts towards reviewing the manuscript and providing encouraging feedback. We did a revision of the manuscript to fix the grammar mistakes and to increase the readability of the paper. |

**Reply to Reviewer #2:**

| | |
|---|---|
| As suggested, the authors have extended the background to include some of the concepts used along the paper. I would also move the definition of online and offline distillation (including figure 2) that now appears at the beginning of section 5 (lines 173-176) to the background. | We thank the reviewer for his/her efforts towards reviewing the manuscript and providing valuable feedback.<br><br>We have updated the background to move the definition of online |

| | and offline distillation including figure 2. |
|---|---|
| The authors have also included a new section summarizing the applications, as requested. This section contributes making the paper more complete.<br><br>New figures are quite useful to understand the background concepts and the categories used to classify the papers.<br><br>The manuscript still contains some grammar mistakes (a few are listed below), so proof-reading is highly recommended before publication.<br><br>- Section 1: The main objectives of this works is => are<br>- Section1: Also, the paper discuss => discusses<br>- Section 3: It's purposes => its purposes<br>- Section 5: the two sub-category => subcategories<br>- Section 6: deep learning models use to run => are usually run ?<br>- Section 6: To be practically in use => To be of practical use<br>- Section 6: To be low latency => To have low latency | We would like to thank the reviewer for providing encouraging feedback.<br><br><br><br>We thank the reviewer for pointing these errors. We fixed them in the updated manuscript. In addition, we proofread the manuscript to fix grammar mistakes elsewhere, as well. |
| I am not fully satisfied with the answer given by the authors about the survey methodology. Even if they do not want to conduct a systematic literature search, the process followed to find and select the papers should be better explained in the manuscript. It seems that the survey is focused on recent works not included in previous surveys, so the covered period of time should be given. The names of the journals and conferences considered as "relevant", as well as the minimum citation count, should be reported as well. Even though these criteria might not be valid for a systematic review, the reader has the right to know how the authors choose papers. Otherwise, the "overview" of the area is strongly biased by the authors' interest on certain papers, but the reader is not aware of it. | Based on the feedback from the reviewer, we have updated the survey methodology by adding details on the paper selection criteria including the publication year and the venue. |
| The new organization of the survey section has greatly contributed readability.<br>The authors have successfully addressed my comments about the validity of findings. | We would like to thank the reviewer for his/her efforts towards reviewing the manuscript and providing encouraging feedback. |